# Impact of COVID-19 on English Football Premier League: Analyzing rankings and home advantage using extended Bradley-Terry models

**Hong Hong Liu, Jack Battaglia, Tong Tong Wu**[ID]*

Department of Biostatistics and Computational Biology, University of Rochester Medical Center, Rochester, New York, United States of America

* tongtong_wu@urmc.rochester.edu

**Data availability statement:** The data underlying the results presented in the study are

## Abstract

COVID-19 significantly impacted sports tournaments, particularly soccer. During the pandemic, restrictions on spectators and frequent team composition changes due to outbreaks, providing a unique opportunity to explore the effects on team performance and home advantage (HA). In this study, we have two focuses. We first introduce a new extended Bradley-Terry model. The proposed model demonstrated several advantages over existing models, particularly in terms of stability, ease of implementation, and interpretation. It can be applied to any sport featuring home and away dynamics. The second focus is to apply the proposed model to data from the English Premier League over the past ten seasons. The analysis aims to evaluate changes in team rankings and home advantage before, during, and after the pandemic disruptions. Our findings indicate marked fluctuations in team performance and home advantage during the pandemic, with a distinct shift in team dynamics and competitive balance under pandemic restrictions.

## Introduction

The English Premier League (EPL) is widely regarded as world's most prestigious soccer league, attracting top talent and substantial financial investment [1]. Its high level of play and passionate fan support create a pronounced home advantage (HA) [2]. However, during the pandemic, the 2019-2020 season was temporarily paused in early March and was resumed in mid-June, with the remaining matches to be played without fans. One significant change that has been adopted was the limiting or prohibiting of fans from attending matches. In addition to matches without fans, teams have been beset by clusters of COVID-19 cases, leading to inconsistent team lineups. The double impact of absent fans and health risks has the potential to disrupt team rankings and diminish the traditional home advantage. Studies, such as McCarrick et al. [3], demonstrate a reduction in HA during crowdless games, attributing it to fewer favorable referee decisions and reduced player motivation. Leitner et al. [4] corroborated these findings, highlighting diminished HA across European leagues during

available from https://www.kaggle.com/code/ameetdhokia/epl-matches-93-22.

**Funding:** This work was supported in part by the National Institute of Dental and Craniofacial Research of the National Institutes of Health (NIH/NIDCR, Grant R01DE031025, PI: Tong Tong Wu, https://www.nidcr.nih.gov/), and by the Smart and Connected Communities Program of the National Science Foundation (NSF SCC, Grant 2238208, Tong Tong Wu, https://www.nsf.gov/funding/pgm_summ.jsp?pims_id=505364). The funders had no role in study design, data collection and analysis, decision to publish, or preparation of the manuscript.

**Competing interests:** The authors have declared that no competing interests exist.

"ghost games," linking it to the psychological effects of absent fans on referees and players alike. Similarly, Wang and Qin [5] confirmed that crowd absence significantly reduces team performance, impacting tactical and physical metrics such as pass completion and shot accuracy. Basini et al. [6] explored EPL competitiveness through a stochastic block model, revealing structural changes in competitive balance that could interplay with pandemic effects on team performance. However, not all evidence aligns with this trend. Scelles et al. [7] reported that home advantage persisted even in fanless matches, suggesting factors beyond crowd presence.

To measure changes in pandemic, we focus on two characteristics: the predictive power of overall team performance and home advantage. Historically, attempts to predict the outcome of soccer games have often involved independently modeling each team's scoring potential and these predictions typically used independent Poisson distributions to predict goals scored. For example, the paper of Maher [8] uses a Poisson model with team-specific offensive and defensive parameters. However, these models are somewhat "static" in the sense that the information used for calculating the strength parameters didn't account for time. Dynamic models were subsequently developed, such as in Dixon and Coles [9] the significance of results was weighted such that the most recent results had stronger impacts on team strength than results from long ago.

Beyond the Poisson modeling of the goals scored, other approaches involve using match-specific data, or even just solely match results such as win/draw/loss to build paired-comparison models to predict the outcome of a game. Kuk [10] proposed a linear paired comparison model for sports with frequent and variable draw rates, where maximum likelihood estimation is not feasible, instead matching observed and expected results to handle home advantage. Schauberger et al. [11] used extensive match-specific data such as ball possession, pass completion rates, distance run by players, and much more to try to build a model capable of outperforming bookmakers at predicting outcomes of matches. Although successful, these were quite complicated models with excessive amounts of parameters reliant on being fed large quantities of match-specific data. Additionally, Schauberger's [11] model uses team-specific intercepts, requiring penalization due to the large number of parameters. While this prevents overfitting, it reduces interpretability, making it difficult to draw clear conclusions about each team's relative advantage and limiting the model's practical value for understanding HA changes. Cattelan et al. [12] alternatively tried to build a model to predict match outcomes using only previous match results (win/draw/loss).

All these models account for home advantage (HA), a phenomenon explained by crowd influence on player motivation and refereeing [13,14], travel fatigue for away teams [15], and venue familiarity [16]. Leitner et al. [4] provided a concise overview of these mechanisms from socio-psychological and environmental perspectives. Empirically, Pollard's analysis confirmed that football shows one of the largest HA effects among major team sports [2]. A comprehensive review by Nevill and Holder [17] summarizes these mechanisms.

In recent years, a growing number of studies have explored machine learning and advanced statistical models to improve predictive accuracy in football match outcomes. Groll et al. [18,19] proposed hybrid methods that combine expert-crafted features with random forests or Poisson regression for international tournament prediction. Similarly, Schauberger and Groll [20] used ensemble models incorporating team rankings and bookmaker odds. Yeung et al. [21] evaluated deep learning and gradient boosting approaches, and highlighted the importance of engineered features such as strength ratings. Machine-learning ranking models have demonstrated strong predictive performance, but these methods demand

large datasets and sacrifice interpretability, making them unsuitable for our relatively small COVID-period data. Constantinou and Fenton [22] introduced the pi-rating, a dynamic extension of the Bradley–Terry model that incorporates score margins and outperforms standard Elo ratings in forecasting English Premier League matches. Macrì Demartino et al. [23] recently proposed a Bayesian Bradley–Terry–Davidson model for international competitions, demonstrating improved performance over Elo-style and goal-based models.

Among rating systems, Elo-type models [24] are widely used due to their simplicity and sequential updating. However, they require calibration of $K$-factors and often lack transparency in parameter interpretation. Importantly, the BT model uses an ordered logistic regression to handle ternary outcomes (win/draw/loss) directly, whereas Elo methods typically omit draws. Szczecinski et al. [25] showed that FIFA's Elo ranking system approximates an online Bradley–Terry model, and advocated for a more data-driven, hybrid alternative. Honda [26] further argued that the Bradley–Terry (BT) model yields more stable and interpretable strength estimates than Elo, as it avoids path dependence and considers the full history of match results. Additionally, the BT model allows for likelihood-based inference and confidence intervals, which are typically unavailable in standard Elo frameworks. The Davidson model [27] expands the BT model to accommodate ties, which enhances flexibility for sports like football and hockey. While Davidson models allow for draws, they do not accommodate the team-specific HA parameterization our study requires, whereas the BT model flexibly incorporates such structure.

Taken together, these findings support the use of BT-type models as a balanced choice between simplicity, interpretability, and empirical performance. Our model omits match-specific covariates and team-specific intercepts, yielding a parsimonious model that requires no penalization and remains interpretable, remains advantageous for (i) handling incomplete pairwise comparisons, (ii) robustness under abrupt disruptions (e.g. COVID-19), (iii) straightforward incorporation of HA parameters, and (iv) parsimonious parameterization.

Utilizing data from the most recent 10 seasons of the English Premier League, we develop and apply an extended Bradley-Terry model and compare it with two existing extended Bradley-Terry models in Cattelan et al. [12]: static and dynamic models. Our work is close related to the one in Schauberger et al. [11]. However, without team-specific intercepts and match-specific covariates, no penalization is needed, therefore resulting in a simpler, more interpretable model. We first examine if the three models can adequately handle the unexpected changes following the COVID-19 pause and continue to perform well. The comparison is made to assess if the proposed model would outperform the static and dynamic models under the situation that matches played after the pause would be so variable and a more flexible model would do better. The three models will be used to fit the data from three different periods. Special attention was given to the evolution of the HA parameter and rank changes estimated using these models, to assess the importance of HA in predicting the outcome of matches.

To address the two focuses, introducing the new Bradley-Terry model and examining the impact of COVID-19 on sports performance, the paper is organized as follows. Methods describes two existing Bradley-Terry models and introduces a new Bradley-Terry model. Results describes the data of ten English Premier League teams over ten seasons and presents the results of applying the three models, as well as the validation and comparison of the three models. The paper is wrapped up with a discussion in Discussion.

## Methods

### Models

Considering each game as a paired comparison of two teams, Bradley and Terry [28] developed the original Bradley-Terry model to evaluate paired matchups of teams in a group,

$$P(Y_i = y_i) = \frac{e^{a_{h_i} - a_{v_i}}}{1 + e^{a_{h_i} - a_{v_i}}}, \quad y_i \in \{0, 1\}, \tag{1}$$

where $a_{h_i}$ and $a_{v_i}$ are parameters that measure the skill level of the team $h_i$ and team $v_i$ in the group. Note that in this original formulation, we are only able to model the probability of a binary outcome ($y_i = 1$ for win or 0 for loss) between team $h_i$ winning against team $v_i$.

### Model 1 – Static Bradley-Terry model with a uniform home advantage factor

To account for ties and home and away field factors, which occur in soccer matches, the original Bradley-Terry model has been extended in different ways. Considering the advantage of playing at home, the first model we consider includes a common home effect parameter $\eta$ for all teams. To accommodate the three possible outcomes (2 for a home team win; 1 for a tie; and 0 for an away team win), the Bradley-Terry model was extended to account for the home advantage parameter as well as ties through a cumulative link specification in Cattelan et al. [12]. For game $i$, it is written as

$$\Pr(Y_i \leq y_i) = \frac{e^{\delta_{y_i} + \eta + a_{h_i} - a_{v_i}}}{1 + e^{\delta_{y_i} + \eta + a_{h_i} - a_{v_i}}}, \quad y_i = \{0, 1, 2\}, \tag{2}$$

where $y_i$ takes a value of 2, 1, or 0 if the home team wins, draws, or loses to the visiting team, respectively and $h_i$, $v_i$ represents the home team and visiting team at match $i$ with $h_i, v_i = 1, ..., n$, $h_i \neq v_i$. Each team has their own skill parameter $a_k$, $k \in \{1, ..., n\}$. A universal home advantage (HA) parameter $\eta$ represents the average HA of all teams in the analysis, and $-\infty < \delta_0 < \delta_1 < \delta_2 = \infty$ are parameters corresponding to the three outcomes. By imposing the 'symmetrical' constraints such that $\delta_0 = -\delta_1 = \delta$, we ensure that any two teams against each other with the same skill parameter and without HA ($\eta = 0$) would have the same probabilities of winning such that

$$\Pr(Y_i = 0) = \frac{e^{\delta_{y_i} + \eta + a_{h_i} - a_{v_i}}}{1 + e^{\delta_{y_i} + \eta + a_{h_i} - a_{v_i}}} = 1 - \Pr(Y_i \leq 1) = \Pr(Y_i = 2).$$

If the number of possible outcomes reduces to Eq (2) (i.e., no ties), $\eta = 0$, and $\delta_{y_i} = 0$, the cumulative logit model Eq (2) reduces to the standard Bradley-Terry model Eq (1). Because of identifiability, a constraint is needed for the capability parameters, such as the sum constraint $\sum_{k=1}^{10} a_k = 0$ or the reference team constraint $a_k = 0$ for team $k \in \{1, ..., n\}$.

### Model 2 – Dynamic Bradley-Terry model

Cattelan et al. [12] proposed a dynamic Bradley-Terry model. Instead of assuming a constant ability parameter for each team over time, the dynamic model allows individual ability parameters to vary over time. The dynamic model is written as the following:

$$\Pr(Y_i \leq y_i) = \frac{e^{\delta_{y_i} + a_{h_i}(t_i) - a_{v_i}(t_i)}}{1 + e^{\delta_{y_i} + a_{h_i}(t_i) - a_{v_i}(t_i)}}, \quad y_i \in \{0, 1, 2\}. \tag{3}$$

Instead of a constant HA parameter $\eta$, the time-varying skill parameters $a_{h_i}(t)$ and $a_{v_i}(t)$ incorporate the HA factor. That is, home team $h_i$'s skill in the current game is calculated as a combination of its HA and its skill in the previous game, such that

$$a_{h_i}(t_i) = \lambda_1 \mu_{h_i}(t_i) + (1 - \lambda_1) a_{h_i}(t_i - 1), \tag{4}$$

where $\lambda_1 \in [0, 1]$ is the home team's smoothing parameter determining the weight given to HA, which comes from $\mu_{h_i}(t_i)$ calculated as

$$\mu_{h_i}(t_i) = \beta_1 r_{h_i}(t_i^{-1}), \tag{5}$$

where $\beta_1$ is the home-specific parameter and $r_{h_i}(t_i^{-1})$ accounts for the number of points earned by home team $h_i$ in the game played on matchday $t_i$. Both $\lambda_1$ and $\beta_1$ need to be estimated by MLE.

With this formula, the HA factor is calculated as an exponentially weighted moving average (EWMA) using all of the home team $h_i$'s previous home results to determine how well they perform at home. We further assume that all the teams start with the uniform HA $\bar{r}_h$, which is the average home performance for all teams in the previous season. Likewise, visiting team $v_i$'s skill can be calculated similarly as in Eq (4) and Eq (5) with the smoothing parameter $\lambda_2$ and visitor-specific parameter $\beta_2$ from the second EWMA process. These parameters could potentially offer more flexibility in estimating HA, especially if there is a significant change in the team's performances in home games as the season goes on.

## Proposed model - Bradley-Terry model with distinct home advantage

The model we propose in this paper addresses the heterogeneity in the home advantage across soccer teams. In fact, some teams are renowned for their dominant home records. For example, Manchester United has won 61.43% of their home fixtures since the 1892–93 season [29], which often attributed to their vibrant supporters and the long history of the Old Trafford stadium. For each team, the model allows for different home and away ability parameters, therefore yielding a distinct HA factor for each team. This setting reflects the fact that the impact of HA is not uniform and varies according to the characteristics and circumstances of each club.

While it shares some similarities with the approach in Schauberger et al. [11], this model is designed specifically to capture each team's unique home advantage (HA) more accurately and with greater interpretability. By discarding team-specific intercepts and match-specific covariates, we avoid the need for penalization, resulting in a simpler, more interpretable model. Thus, the proposed model is given by

$$\Pr(Y_i \leq y_i) = \frac{e^{\delta_{y_i} + a_{h_i}^H - a_{v_i}^V}}{1 + e^{\delta_{y_i} + a_{h_i}^H - a_{v_i}^V}}, \quad y_i = \{0, 1, 2\}, \tag{6}$$

where $-\infty < \delta_0 < \delta_1 < \delta_2 = \infty$. It can be seen from the model that each team has its own home ability parameter $a_k^H, k \in \{1, ..., 10\}$ and visiting ability parameter $a_k^V, k \in \{1, ..., 10\}$. The difference between a team's home and away ability parameters

$$\eta_k = a_k^H - a_k^V \tag{7}$$

yields the HA of each team. In this model, we only need to set one constraint about the visiting ability of the reference team $a_k^V = 0$ for team $k \in \{1, ..., n\}$.

### Likelihood inference

The parameters of the dynamic Bradley-Terry model were estimated using maximum likelihood estimation (MLE). Given the observed match outcomes, we maximized the following likelihood function with respect to the model parameters:

$$\mathcal{L}(\theta; \mathbf{y}) = \prod_{i=1}^{n} \Pr(Y_i = y_i \mid Y_{i-1} = y_{i-1}, ..., Y_1 = y_1; \theta), \qquad (8)$$

where $\theta = (\beta_1, \beta_2, \lambda_1, \lambda_2, \delta)^\top$ includes the home- and away-specific coefficients $(\beta_1, \beta_2)$, the smoothing parameters $(\lambda_1, \lambda_2)$, and the cutpoints $\delta$ used in the cumulative logistic model.

The conditional probability for each match $Y_i$ (home win, draw, or away win) is modeled as:

$$\Pr(Y_i \leq y_i) = \frac{\exp\{\delta_{y_i} + \beta_1 x_{h_i}(t_i; \lambda_1) - \beta_2 x_{v_i}(t_i; \lambda_2)\}}{1 + \exp\{\delta_{y_i} + \beta_1 x_{h_i}(t_i; \lambda_1) - \beta_2 x_{v_i}(t_i; \lambda_2)\}}, \quad y_i \in \{0, 1, 2\}, \qquad (9)$$

where $x_{h_i}(t_i; \lambda_1)$ and $x_{v_i}(t_i; \lambda_2)$ are exponentially weighted moving averages (EWMA) of past home and away performances, respectively, defined as:

$$x_{h_i}(t_i; \lambda_1) = \lambda_1 \sum_{k=0}^{K-1} (1 - \lambda_1)^k r_{h_i}(t_i^{(-k-1)}) + (1 - \lambda_1)^K \bar{r}_h, \qquad (10)$$

$$x_{v_i}(t_i; \lambda_2) = \lambda_2 \sum_{k=0}^{K-1} (1 - \lambda_2)^k r_{v_i}(t_i^{(-k-1)}) + (1 - \lambda_2)^K \bar{r}_a. \qquad (11)$$

Here, $r_{h_i}(t_i^{(-k-1)})$ and $r_{v_i}(t_i^{(-k-1)})$ denote the number of points obtained by the home and away teams in their respective prior matches. The terms $\bar{r}_h$ and $\bar{r}_a$ are the average number of points obtained at home and away, respectively, across all teams in the previous season, and serve as baseline initializations.

The smoothing parameters $\lambda_1$ and $\lambda_2$ were determined via profile likelihood maximization, allowing the model to automatically tune the balance between historical performance and recent form. This specification closely follows the dynamic Bradley-Terry model proposed by Cattelan et al. [12].

## Results

### Data

The raw dataset contains 3,800 matches in the English Premier League (EPL) from 2014 to 2023, recording the results between the home and away teams among a total of 29 teams. Considering the league's relegation rules, only ten teams remained consistent throughout the 10 years. Focusing on these ten teams minimizes the variability caused by teams with inconsistent league presence, as some teams may perform exceptionally well in one season but not appear often in others, such as Leicester City in the 2015-2016 season. Including such teams could skew the relative ranking of team strength across seasons. Therefore, this study analyzes only the teams that have been in the Premier League throughout the entire period, resulting

in a dataset of 900 matches with consistent home and away records, focusing on the relative rankings of teams with stable EPL presence. Importantly, before the 2019–2020 season, the Video Assistant Referee (VAR) system was introduced in the EPL, which has been shown to affect match outcomes and may potentially mitigate home-team refereeing bias [30,31]. The ten seasons of 2014-2023 can be divided into three periods: pre-COVID (2014-2019), during COVID (2020-2022), and post-COVID (2023). To evaluate the performance of the models, the data in each of the three periods are further divided into training and testing sets: data from 2014 to 2018 as training and data in 2019 as testing for the pre-COVID period; data from 2020 to 2021 as training and data in 2022 as testing during COVID; and data in 2023 will be used as training only for post-COVID since only one year of data are available. The models are fitted using m likelihood method. All model parameters were estimated by maximizing the likelihood function using the likelihood `nlminb` optimizer in R.

## Results from static model using training data

**Rank changes using static model.** Table 1 shows that the teams' rankings fluctuated either up then down or down then up, while the overall trend was for very little change. This suggests that the pandemic affected the performance of the teams differently. Manchester City's team remained on top all the time. The team's performances, even during the pandemic, suggest that the lack of crowds has had minimal negative impact. This may be due to a style of play that emphasizes ball control and tactical discipline, traits that are less influenced by crowd support. Arsenal's improvement to third place in the post-COVID rankings may be attributed to their younger players, who may have helped them recover more quickly and regain their fitness levels after the pandemic disruption. Notably, Arsenal were the youngest Premier League team in 2021–22 with an average starting age of 24 years 308 days [32] and the second-youngest in 2022–23 with an average starting age of 25 years 52 days [33]. Younger athletes often exhibit greater physiological resilience and faster recovery from fatigue and injury, supporting this interpretation [34].

**Changes in home advantage using static model.** The home advantage (HA) factors estimated by the static model before, during, and after COVID are reported in Table 2. It shows a significant decrease during the pandemic (0.1519) compared to before COVID (0.3708), which is consistent with the absence of spectators in the stadium. This supports the idea that

**Table 1**. Estimated ranks by the static model and actual rank using training data.

| Team | Pre-COVID | | | | | COVID | | | | | Post-COVID | | | | |
|---|---|---|---|---|---|---|---|---|---|---|---|---|---|---|---|
| | Model | | Actual Rank | | | Model | | Actual Rank | | | Model | | Actual Rank | | |
| | Skill Est | Rank | A | B | C | Skill Est | Rank | A | B | C | Skill Est | Rank | A | B | C |
| Manchester City | 0.275 | 1 | 1 | 1 | 3 | -0.450 | 2 | 1 | 1 | 2 | 1.230 | 1 | 1 | 1 | 1 |
| Chelsea | 0.251 | 2 | 4 | 2 | 2 | -1.202 | 4 | 4 | 4 | 4 | -1.123 | 6 | 7 | 7 | 6 |
| Manchester Utd | 0.084 | 3 | 6 | 6 | 1 | -1.019 | 3 | 3 | 3 | 3 | 0.293 | 2 | 3 | 3 | 3 |
| Liverpool | 0.000 | 4 | 5 | 5 | 4 | 0.000 | 1 | 2 | 2 | 1 | 0.000 | 4 | 4 | 4 | 4 |
| Tottenham | -0.090 | 5 | 2 | 4 | 5 | -1.366 | 7 | 5 | 5 | 7 | -0.430 | 5 | 5 | 5 | 5 |
| Arsenal | -0.132 | 6 | 3 | 3 | 6 | -1.320 | 6 | 6 | 6 | 5 | 0.278 | 3 | 2 | 2 | 2 |
| Everton | -0.777 | 7 | 7 | 7 | 8 | -1.311 | 5 | 7 | 7 | 6 | -1.190 | 7 | 9 | 9 | 7 |
| Southampton | -0.864 | 8 | 8 | 8 | 9 | -2.104 | 9 | 9 | 9 | 8 | -1.376 | 8 | 10 | 10 | 9 |
| West Ham | -0.934 | 9 | 9 | 9 | 7 | -1.989 | 8 | 8 | 8 | 9 | -1.752 | 10 | 8 | 8 | 10 |
| Crystal Palace | -1.407 | 10 | 10 | 10 | 10 | -2.173 | 10 | 10 | 10 | 10 | -1.423 | 9 | 6 | 6 | 8 |

*Note:* In the actual world, rank A represents the mean rank in EPL using the 0-1-3 scoring system, rank B represents the mean rank in EPL using the 0-1-2 scoring system, and rank C represents the mean rank in EPL using the 0-1-2 scoring system and filtered data (only the 10 teams).

**Table 2. Estimated uniform HA by the static model using training data.**

|  | Pre-COVID | COVID | Post-COVID |
|---|---|---|---|
| Home Advantage | 0.3708 | 0.1519 | 0.8504 |

*Note:* HA is the estimated average home advantage under the static model for each period.

spectator support has been a major factor in home advantage in soccer. As the pandemic abated and the fans returned, the post-COVID HA increased significantly (0.8504), likely due to the return of the fans' influence and the referee's preference towards the home team.

## Results from dynamic model using training data

Since the likelihood function involves averaging data from the preceding year and only one year of data (2023) are available post-COVID, the dynamic modeling will focus exclusively on the pre-COVID and during-COVID periods.

Table 3 shows that the estimated smoothing parameters and specific coefficients changed for home and away teams before and during COVID. The increase in smoothing parameters $\lambda_1$ and $\lambda_2$ during the COVID period (0.1736 and 0.3752 compared to pre-COVID values of 0.0025 and 0.0014) suggests that the model heavily relies on prior match outcomes to assess current abilities. However, given the significant shifts in team performance dynamics during the pandemic, the pre-COVID results are not reliable indicators of performance during COVID. This reliance undermines the model's effectiveness, as it cannot adequately adjust to the abrupt changes caused by the pandemic.

During the pre-COVID period, the trends in home and away abilities were stable, maintaining a consistent difference that reflects a clear and steady home advantage. As shown in Fig 1, home and away abilities follow similar trajectories with home ability consistently higher, reinforcing the traditional home advantage effect.

However, in the COVID period (Fig 2), the home and away abilities often intersect, indicating that the home advantage was no longer distinct and may have even turned negative at times. This suggests that the home advantage diminished significantly during the pandemic, likely due to the lack of spectators. Yet, the frequent crossing of these ability lines could also indicate limitations in the model's ability to accurately capture these changes in an unusual context. The model's reliance on prior match data and increased smoothing parameters may fail to adjust adequately to the unprecedented shift in team dynamics during COVID, leading to potential inaccuracies in estimating true team abilities.

The home-specific parameter $\beta_1$ decreases dramatically from -16.3320 in the pre-COVID period to -0.7128 during the COVID period, and conversely, the away-specific parameter $\beta_2$ changes significantly from -25.7924 to 0.5478. Negative values in the pre-COVID period

**Table 3. Estimated parameters by the dynamic model using training data.**

|  |  | Pre-COVID | COVID |
|---|---|---|---|
| Smoothing Parameters | $\lambda_1$ | 0.0025 | 0.1736 |
|  | $\lambda_2$ | 0.0014 | 0.3752 |
| Home-specific parameter | $\beta_1$ | -16.3320 | -0.7128 |
| Away-specific parameter | $\beta_2$ | -25.7924 | 0.5478 |

*Note:* $\lambda_1$ and $\lambda_2$ are smoothing parameters; $\beta_1$ and $\beta_2$ represent home-specific and away-specific coefficients, respectively.

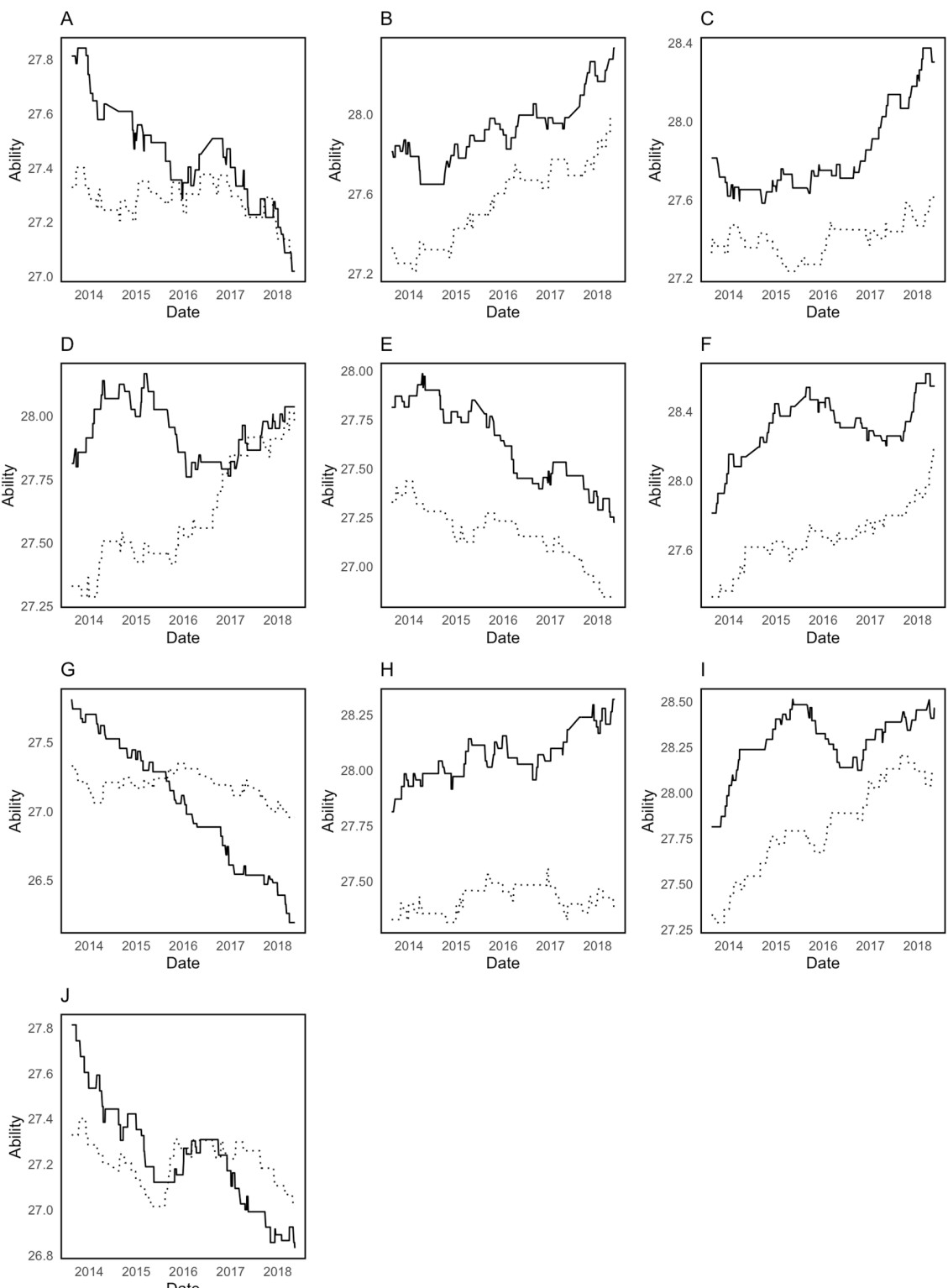

**Fig 1. Smoothed home (—) and away (………) ability parameters for 10 teams before COVID.** A Southampton; B Manchester Utd; C Tottenham; D Liverpool; E Everton; F Manchester City; G Crystal Palace; H Arsenal; I Chelsea; J West Ham.

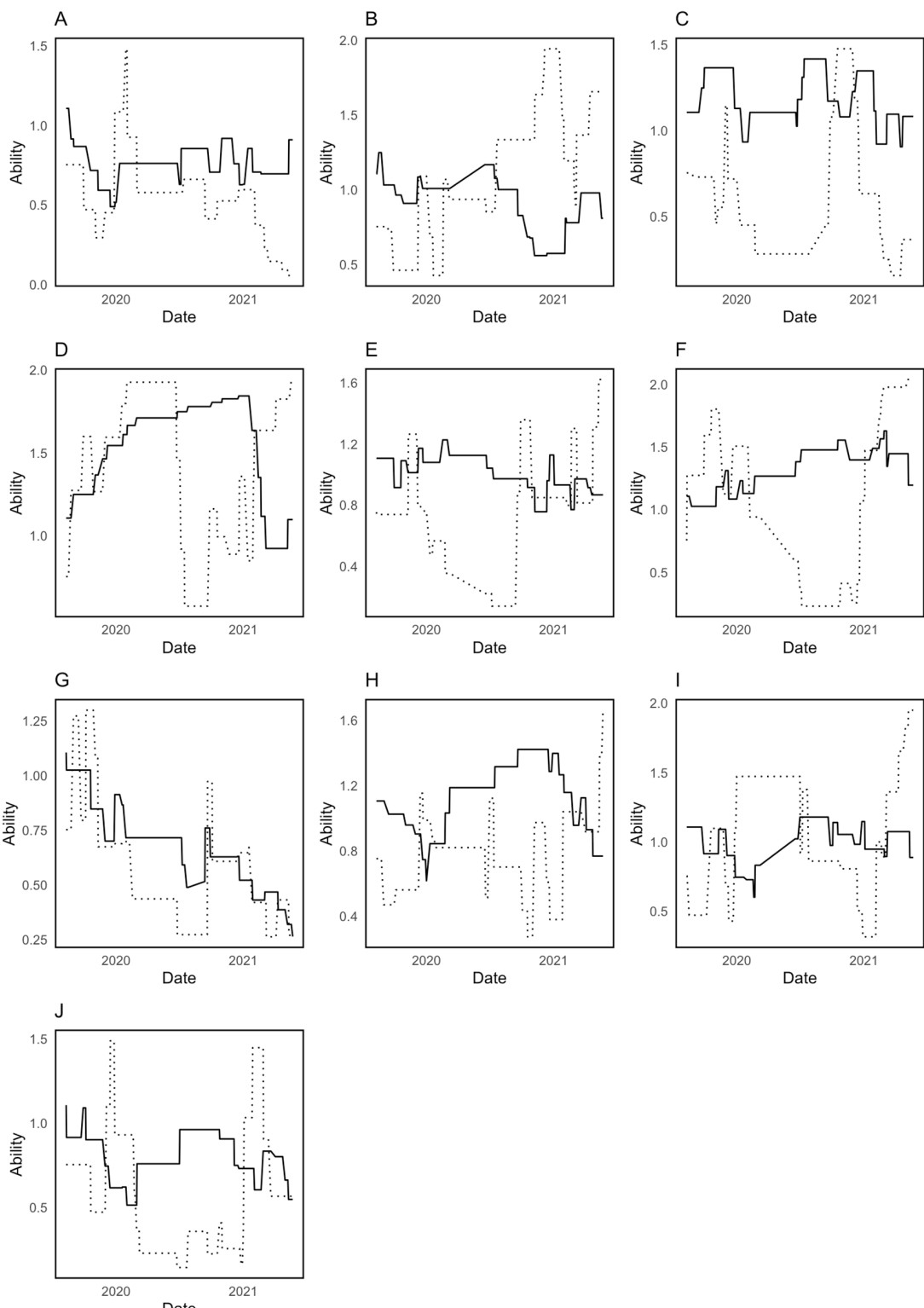

**Fig 2. Smoothed home (—) and away (………) ability parameters for 10 teams during COVID.** A Southampton; B Manchester Utd; C Tottenham; D Liverpool; E Everton; F Manchester City; G Crystal Palace; H Arsenal; I Chelsea; J West Ham.

may indicate that the observed effect of home-away disadvantage is much stronger, whereas it shifts to a near-neutral effect during the COVID period. This is consistent with the static model's change in HA, whereby the home advantage diminishes in the absence of spectators, thus neutralizing the traditional home-away effect. Nevertheless, using a single HA for all teams throughout different periods is a coarse approach that overlooks the nuanced reality of varying impacts on different clubs, suggesting the need for a better model that considers individual team factors to provide a more realistic representation.

## Results from proposed model using training data

Rank Changes Using Proposed Model

By analyzing the overall trend of the home and away ability rankings of the ten English Premier League teams over the three periods of time in Table 4, we observe their synchronized changes, i.e., the rankings either rise and then fall, or fall and then rise. For most teams, the decline in home ability rankings from the pre-COVID period to the COVID period is usually more pronounced than the decline in away ability rankings. This pattern may reflect the effect of the absence of home crowds during the epidemic period, which has historically been a key factor in home performance. The transition from the COVID period to the post-COVID period was characterized by a general rise in home capacity rankings, which tended to be greater compared to away capacity. This may represent a resurgence of home advantage as fans return to the stadium and teams readjust to the familiar atmosphere of playing with home support.

For example, Manchester United was ranked 4th in home ability and 4th in away ability, with a balanced home and away strength before COVID. During COVID-19, their home ability ranking dropped dramatically to 7th place, more so than their away ranking which dropped to 5th place, which suggests that the lack of home crowds has had a more pronounced effect on their performance, as Old Trafford is known for its strong home support. This overall pattern emphasizes the important impact of crowds on home performance.

**Table 4. Estimated ranks by the proposed model and actual ranks using training data.**

| Team | Pre-COVID | | | | COVID | | | | Post-COVID | | | |
|---|---|---|---|---|---|---|---|---|---|---|---|---|
| | Pred. Home Ability Rank | Actual Home Victory Mean Rank | Pred. Away Ability Rank | Actual Away Victory Mean Rank | Pred. Home Ability Rank | Actual Home Victory Mean Rank | Pred. Away Ability Rank | Actual Away Victory Mean Rank | Pred. Home Ability Rank | Actual Home Victory Mean Rank | Pred. Away Ability Rank | Actual Away Victory Mean Rank |
| Manchester City | 1 | 1 | 2 | 1 | 2 | 1 | 3 | 1 | 2 | 1 | 1 | 2 |
| Chelsea | 2 | 3 | 1 | 2 | 6 | 4 | 4 | 4 | 7 | 8 | 4 | 6 |
| Arsenal | 3 | 2 | 6 | 6 | 5 | 6 | 5 | 6 | 3 | 3 | 2 | 1 |
| Manchester Utd | 4 | 5 | 4 | 5 | 7 | 5 | 2 | 3 | 1 | 2 | 6 | 3 |
| Tottenham | 5 | 4 | 5 | 3 | 3 | 3 | 8 | 9 | 5 | 5 | 8 | 4 |
| Liverpool | 6 | 5 | 3 | 4 | 1 | 2 | 1 | 2 | 4 | 4 | 5 | 4 |
| Everton | 7 | 6 | 9 | 9 | 4 | 8 | 6 | 5 | 8 | 8 | 9 | 10 |
| Southampton | 8 | 7 | 8 | 7 | 9 | 8 | 10 | 7 | 10 | 9 | 3 | 7 |
| West Ham | 9 | 8 | 7 | 10 | 8 | 7 | 9 | 7 | 6 | 6 | 10 | 9 |
| Crystal Palace | 10 | 9 | 10 | 8 | 10 | 10 | 7 | 10 | 9 | 7 | 7 | 2 |

*Note:* Actual home victory mean rank represents the rank of the percentage of victories in home-field matches; similarly, actual away victory mean rank is defined as such.

**Changes in home advantage using proposed model.** By setting the reference team 'Liverpool' ability parameter as 0, we can calculate the home ability parameter and away ability parameter of each team, which would give us the home advantage of every team from the difference of these two parameters. In this case, the change of HA in the reference team could not be identified as it is zero all the way, however, there is marked meaning in the change of each team's HA rank. Table 5 indicates that clubs such as Manchester United, Arsenal, and Chelsea—whose strong fan bases are reflected in their historically high average home attendances (56, 12, and 10 seasons leading the league, respectively) [35]—experienced a significant drop in HA during the pandemic. In post-COVID, the trends diverge. Chelsea's continued decline implies a significant lasting impact of COVID on their HA, indicating they are still seeking strategies to recover.

Arsenal and Manchester United show HA rank increases post-COVID, possibly due to their historical resilience and adaptation to the new normal, regaining strong home-field support. On the contrary, Liverpool and Manchester City witnessed an increase during COVID, since they are strong enough whether at home or visiting field. Other clubs with less fan support experienced an increase during COVID-19 and a decrease after the pandemic, which could be because the teams with higher HA ranks were more affected by the absence of fans, leveling the playing field to some extent.

## Model comparison

To compare the performance of the above models, we employ the following three criteria. The Brier Score (BS) is a measure of the accuracy of probabilistic predictions. It quantifies the difference between predicted probability and the actual outcome. Lower Brier Scores indicate more accurate predictions, making it a robust tool for assessing the precision of our models' forecasts. The Brier Score for match $i$ is calculated by

$$BS_i = \sum_{y_i=0}^{2} \left[ \Pr(Y_i = y_i) - I(Y_i = y_i) \right]^2, \quad y_i = \{0, 1, 2\}. \tag{12}$$

Rank Probability Score (RPS) evaluates the accuracy of predicted rankings. It considers the probability distribution over all possible rank outcomes and compares it to the actual result. Like the Brier Score, a lower RPS signifies better performance in predicting team ranks.

**Table 5**. HA estimates by the proposed model using training data.

| Team | Pre-COVID | | COVID | | Post-COVID | |
|---|---|---|---|---|---|---|
| | HA | Rank | HA | Rank | HA | Rank |
| Arsenal | 1.0896 | 1 | 0.1675 | 7 | 1.2842 | 5 |
| Tottenham | 0.7502 | 2 | 1.1689 | 1 | 1.7475 | 3 |
| Everton | 0.6797 | 3 | 0.5095 | 4 | 0.0216 | 7 |
| Manchester Utd | 0.4573 | 4 | -0.8898 | 10 | 3.3581 | 1 |
| Manchester City | 0.3891 | 5 | 0.6280 | 2 | 1.1556 | 6 |
| Chelsea | 0.3021 | 6 | -0.4864 | 8 | -0.4929 | 9 |
| Southampton | 0.1809 | 7 | 0.5035 | 5 | -1.4267 | 10 |
| Liverpool | 0.1614 | 8 | 0.5521 | 3 | 1.7238 | 4 |
| West Ham | -0.0214 | 9 | 0.3047 | 6 | 3.0813 | 2 |
| Crystal Palace | -0.4907 | 10 | -0.6895 | 9 | 0.3151 | 8 |

*Note:* Home Advantage (HA) is calculated as the difference between home and away ability parameters. Liverpool was set as the reference team.

$$RPS_i = \sum_{y_i=0}^{2} \left[ \Pr(Y_i \le y_i) - I(Y_i \le y_i) \right]^2, \quad y_i = \{0, 1, 2\}. \tag{13}$$

The Akaike Information Criterion (AIC) is used to measure the quality of a model in terms of the trade-off between the goodness of fit and the simplicity of the model. A model with a lower AIC is considered more efficient as it provides a better fit with fewer parameters. It is given by

$$AIC = 2k - 2 \ln L, \tag{14}$$

where $k$ is the number of parameters in the statistical model, and $L$ is the model likelihood given the data.

Using these criteria, we compare the prediction performance of the three models to the testing data. The comparison focuses on the model's ability to provide accurate predictions and efficient parameterization in reflecting the changes in the pre-COVID (2019) and COVID (2022) periods. The post-COVID period is not considered because only one year of data are available in this period. Table 6 shows that the proposed model has satisfactory performance in predicting football match outcomes in the English Premier League. It has the lowest mean Brier Score (0.184) and mean Rank Probability Score (0.190) in the pre-COVID period, indicating it was the most accurate in forecasting results before COVID, with fewer errors in probability estimations. During COVID, the static model and the proposed model had similar BS and RPS, while the dynamic model worsened slightly (0.212). Additionally, the higher AIC for the proposed model than the static and dynamic models is due to the number of parameters. The dynamic model only has four parameters, and if we compare based solely on negative log-likelihood, our model performs better.

## Model validation using testing data

**Static model.** Table 7 compares the actual ranks of the test dataset across different scoring systems with the model ranks before and during COVID. For successful validation, we would expect the model ranks to closely match the actual ranks, or at least follow the same trend.

Before COVID, the static model appears to be reasonably close to the actual ranks. For example, Manchester City consistently holds top positions in both the model and actual ranks, suggesting that the model's predictions were robust during stable periods. However, there are certain deviations in the model's predictions from the actual ranks during the COVID. For example, Manchester Utd was ranked 3rd by the model pre-COVID but dropped to actual ranks of 6th and 7th in different scoring systems. Post-COVID, the model rank for Manchester Utd was still 3rd, while the actual ranks were 2nd and 4th. This discrepancy suggests that the model did not fully capture the impact of COVID on Manchester Utd's performance.

**Table 6. Model Comparison Criteria using testing data pre-COVID and during COVID.**

|  | Pre-COVID | | | | COVID | | | |
|---|---|---|---|---|---|---|---|---|
|  | BS | RPS | $-\ln L$ | AIC | BS | RPS | $-\ln L$ | AIC |
| Static | 0.188 | 0.195 | 85.964 | 193.928 | 0.205 | 0.214 | 92.333 | 208.836 |
| Dynamic | 0.232 | 0.261 | 102.514 | 215.027 | 0.212 | 0.229 | 95.030 | 202.006 |
| Proposed | 0.184 | 0.190 | 84.472 | 208.944 | 0.206 | 0.216 | 93.471 | 226.943 |

*Note:* Lower values of Brier Score (BS), Ranked Probability Score (RPS), $-\ln L$, and AIC indicate better model performance.

**Table 7. Estimated ranks by the static model and actual rank using testing data.**

| Team | Pre-COVID | | | | COVID | | | |
|---|---|---|---|---|---|---|---|---|
| | Model | Actual Rank | | | Model | Actual Rank | | |
| | | A | B | C | | A | B | C |
| Manchester City | 1 | 1 | 2 | 1 | 2 | 1 | 1 | 1 |
| Chelsea | 2 | 3 | 3 | 3 | 4 | 4 | 4 | 3 |
| Manchester Utd | 3 | 6 | 6 | 7 | 3 | 2 | 2 | 4 |
| Liverpool | 4 | 2 | 1 | 2 | 1 | 3 | 3 | 2 |
| Tottenham | 5 | 4 | 5 | 8 | 7 | 6 | 6 | 7 |
| Arsenal | 6 | 5 | 4 | 5 | 6 | 7 | 7 | 6 |
| Everton | 7 | 7 | 7 | 6 | 5 | 8 | 8 | 5 |
| Southampton | 8 | 10 | 10 | 9 | 9 | 10 | 10 | 9 |
| West Ham | 9 | 8 | 8 | 4 | 8 | 5 | 5 | 8 |
| Crystal Palace | 10 | 9 | 9 | 10 | 10 | 9 | 9 | 10 |

*Note:* Rank A is based on the EPL 0-1-3 scoring system, rank B on the 0-1-2 system, and rank C on the 0-1-2 system with filtered data (10 teams only).

When comparing the validation criteria, the increased BS and RPS during COVID (0.205 and 0.214, respectively) relative to Pre-COVID (0.188 and 0.195, respectively) indicate that the model's predictions were less accurate during the pandemic. The increased AIC during COVID (208.836) compared to Pre-COVID (193.928) suggests that the model's fit to the data was less optimal during the pandemic, which may reflect the unpredictability and changes in team dynamics not captured by the model.

**Proposed model.** From Table 8, we can assess the performance of the proposed model by how closely the predicted skill ranks for home and away matches match the actual winning ranks.

Using Liverpool as an example, before COVID their home ability was ranked as 6 by the proposed model and away ability as 3, consistent with their actual home win rank of 5 and away win rank of 4. During COVID, both the predicted home ability and actual home win rank improved significantly to 1 and 2, respectively, indicating that the model accurately captured Liverpool's substantial improvement in performance during this period. This close

**Table 8. Predicted ranks by the proposed model in testing data.**

| Team | Pre-COVID | | | | COVID | | | |
|---|---|---|---|---|---|---|---|---|
| | Home | | Away | | Home | | Away | |
| | Ability Rank | Victory Rank | Ability Rank | Victory Rank | Ability Rank | Victory Rank | Ability Rank | Victory Rank |
| Manchester City | 1 | 1 | 2 | 1 | 2 | 1 | 3 | 1 |
| Chelsea | 2 | 3 | 1 | 2 | 6 | 4 | 4 | 4 |
| Arsenal | 3 | 2 | 6 | 6 | 5 | 6 | 5 | 6 |
| Manchester Utd | 4 | 5 | 4 | 5 | 7 | 5 | 2 | 3 |
| Tottenham | 5 | 4 | 5 | 3 | 3 | 3 | 8 | 8 |
| Liverpool | 6 | 5 | 3 | 4 | 1 | 2 | 1 | 2 |
| Everton | 7 | 6 | 9 | 9 | 4 | 8 | 6 | 5 |
| Southampton | 8 | 7 | 8 | 7 | 9 | 8 | 10 | 7 |
| West Ham | 9 | 8 | 7 | 10 | 8 | 7 | 9 | 7 |
| Crystal Palace | 10 | 9 | 10 | 8 | 10 | 9 | 7 | 9 |

*Note:* Actual home victory ranks are calculated from the points earned by each team in matches played at home. Similarly, actual away victory ranks are calculated from matches played away.

correspondence indicates the strong predictive performance of the model under both typical and pandemic-affected conditions.

Comparing the Brier Score and Ranked Probability Score between the Pre-COVID and COVID periods from Table 6, we see that for the proposed model, both the BS (from 0.184 to 0.206) and the RPS (from 0.190 to 0.216) increase during COVID, indicating a slight decrease in predictive accuracy during this more volatile period. However, if we compare these values with the static model's BS (from 0.188 to 0.205) and RPS (from 0.195 to 0.214), the increase in the proposed model's scores is smaller, indicating that it handled the uncertainty of the COVID period slightly better. Although the AIC value for our model is higher, indicating increased complexity, this does not necessarily compromise the model's accuracy.

*Remark: As the dynamic model involves the previous year's data, it is not appropriate to use the training data during model validation using testing data. Therefore, we don't include the validation of the dynamic model.*

## Testing of home advantage parameters

To test the change between three periods instead of the three stratified models above, we use a general model which extends the original static model to account for different time periods: pre-COVID, during COVID, and post-COVID. In this model, the cumulative probability of the outcome $Y_i$ for match $i$ is given by:

$$\Pr(Y_i \leq y_i) = \frac{e^{\delta_{y_i}+\eta_t+a_{h_i,t}-a_{v_i,t}}}{1+e^{\delta_{y_i}+\eta_t+a_{h_i,t}-a_{v_i,t}}}, \quad y_i = \{0,1,2\}, \tag{15}$$

where $\delta_{y_i}$ are associated with the outcome $Y_i$, $\eta_t$ is the HA parameter for the period $t$ (pre-COVID, COVID, or post-COVID), $a_{h_i,t}$ represents the ability of the home team during period $t$, and $a_{v_i,t}$ represents the ability of the visiting team during the same period.

After fitting the model, we construct linear contrasts to test whether the differences in HA between the periods were statistically significant. The results of the linear contrasts, $\eta_{\text{covid}} - \eta_{\text{pre}}$, $\eta_{\text{post}} - \eta_{\text{pre}}$, and $\eta_{\text{post}} - \eta_{\text{covid}}$, are summarized in Table 9. The *p*-values indicate that the difference in HA between the COVID period and the pre-COVID period is not statistically significant ($p = 0.4072$). The difference in HA between the post-COVID period and the pre-COVID period is 0.3839. However, due to the small sample size of post-COVID data, the difference is not significant but not far from being significant at the 10% level ($p = 0.1133$). The difference between post-COVID and during COVID is 0.5049 with a significant *p*-value 0.0491. The tendency of fitted value of these three parameters are in line with the results of stratified models.

**Table 9. Tests of differences in uniform HA estimated by the static model between periods.**

| Linear Contrast | Estimate | StdError | ZValue | PValue |
|---|---|---|---|---|
| $\eta_{\text{covid}} - \eta_{\text{pre}}$ | -0.1210 | 0.1460 | 0.8289 | 0.4072 |
| $\eta_{\text{post}} - \eta_{\text{pre}}$ | 0.3839 | 0.2425 | -1.5833 | 0.1133 |
| $\eta_{\text{post}} - \eta_{\text{covid}}$ | 0.5049 | 0.2567 | -1.9669 | 0.0491 |

*Note: $\eta_t$ represents the home advantage for each period; p-values are from Wald tests.*

## Discussion

### The effect of COVID-19

The impact of COVID-19 on the English Premier League (EPL) was significant, impacting team performance and home advantage (HA), as analyzed by static, dynamic, and our proposed model. The pandemic caused fluctuations in team rankings, while the HA decreased during COVID-19 and then increased after the pandemic, indicating the return of spectators and their significant role in home advantage. The dynamic model highlighted the disruptive effect of the pandemic, with past performance becoming an unreliable predictor, reflecting the profound and unanticipated effects of COVID-19. This disruption was evident in the reduced home advantage during the pandemic. The proposed model provided detailed team-specific HA analysis, revealing significant drops in home performance for teams with strong home support, such as Manchester United, which has dominated attendance records historically, logging the highest average home attendance in 56 out of 124 seasons on record [35]. Post-COVID recovery varied, demonstrating the resilience of different teams to the challenges of the pandemic. Furthermore, all models performed worse during COVID-19 compared to the other two periods, indicating the unexpected impact of the pandemic.

### Comparison of models

**Rank changes.** The fluctuations in football clubs' rankings during various COVID-19 phases reflect the adaptability and strategic approaches unique to each team. Static, dynamic, and proposed models offer different perspectives on these changes, highlighting both the resilience and vulnerability of clubs during the pandemic.

The static model's rank fitting when compared with Rank A (the actual average rank) shows a greater degree of inconsistency than with Rank C. This could be because Rank A is influenced by the full set of teams in the EPL, not just these 10 teams. During the pre-COVID period, there was a notable discrepancy between the model's predictions and Rank A, likely due to the larger dataset available from this period. In contrast, the COVID and post-COVID periods show no significant difference between the 0-1-2 scoring system used by the model and the 0-1-3 scoring system representing actual ranks.

The dynamic model incorporates team abilities that evolve based on their previous match data. However, the estimated ability parameters for each team using this dynamic model are similar results the model struggling to capture the rank changes during COVID, possibly due to the high uncertainty of this period, which disrupts the continuity of past data trends. The smoothing parameter's approach to zero suggests a collapse back to a static model form, indicating that the dynamic model's reliance on historical data is insufficient to account for the unprecedented changes during COVID. This highlights the need for models that can quickly adapt to sudden shifts in external conditions without relying solely on past trends.

The proposed model's ability to provide different rankings for home and away scenarios represents a significant refinement that better captures the true essence of soccer dynamics. By differentiating between home and away performances, the model recognizes the unique influences and pressures that teams experience in different environments. This distinction is critical because home games often benefit from local support and familiarity with the field, while away games present challenges such as travel fatigue and lack of fan support that can affect team performance. It's not just about overall ability, but also the environment in which games are played, making this model particularly relevant for analyses during and after the pandemic - a period characterized by significant disruptions to normal sporting conditions.

**Home advantage.** In discussing the home advantage (HA) ranks across different periods, it is essential to consider the variability and complexity introduced by COVID-19.

The HA of the static model showed a significant variation over the three periods. Before the COVID period, the HA was relatively high, reflecting that teams tend to perform better when playing in their home stadium, likely due to the familiar environment, local crowd support, and lack of travel fatigue. However, the HA dropped sharply during COVID. This dramatic decline is consistent with the restrictions on attendance and the consequent reduction in the traditional components of HA, such as crowd support and referees' bias. The diminished HA during this period illustrates the impact of the pandemic on the typical benefits of playing at home. After the pandemic, the HA increased significantly, indicating a substantial recovery of the HA. This could be due to fans returning to the stadiums and restoring the usual atmosphere of home games, as well as teams possibly regaining their pre-pandemic form and using HA more effectively after adapting to the changes brought about by the pandemic.

For the dynamic model, the overall HA can be derived by subtracting the away-specific parameter from the home-specific parameter. Since the difference in home and away ability for each team is small (less than 1) during the COVID period compared to the pre-COVID period, it suggests that the HA was considerably weakened during this period. This small difference indicates that the distinction between home and away games was less pronounced, a similar result in the static model.

In addition, the dynamic model parameters did not effectively capture the shifts in HA during COVID. As shown in the figures, each team's HA displayed unusual patterns during this period, often fluctuating significantly, with values frequently becoming very small or even negative. This irregular behavior could be due to the high smoothing parameters during COVID, which indicate a strong reliance on past outcomes. However, since the COVID period was very different from pre-COVID times, this reliance on historical data caused the model to misrepresent the changes in HA. As a result, the dynamic model was unable to accurately reflect the shifts in HA for each team over time, highlighting the limitations of using a model that depends on past results in an environment with sudden disruptions. Our proposed model provides a distinctive evaluation of HA by giving a rank for each team's HA, allowing for a detailed analysis of how each team's HA fluctuates over the three different periods. The benefit of this model is its team-specific HA analysis. This detailed view of individual team HA ranks, which isn't available in the other models, is critical to understanding how different teams may be affected by, or adapt to, changes such as those brought about by the pandemic. Our proposed model's HA performance during COVID is more stable compared to the dynamic model, possibly because it's less dependent on the immediate past, which was highly variable during the pandemic. For further improvement, match-specific covariates could be included, such as the number of players affected by COVID-19, injuries, and the number of fans present for each team during pandemic games. However, due to the difficulty in obtaining this data, these factors were not incorporated in this study.

Our proposed model allows us to track changes in home advantage for individual teams, providing a more detailed perspective. In contrast, the static model summarizes the overall trend across the league. By combining both approaches, we can better understand league-wide shifts as well as team-specific effects of COVID-19 on home advantage. This integrated approach would provide a comprehensive understanding of HA dynamics, both at the league level and within individual team contexts.

## Limitation and future research

Although our Bradley-Terry model demonstrates robustness and adaptability, there are several areas for future development. First, future studies could apply our stable model across different leagues or similar competitive contexts to validate its broader utility. Additionally, extending the analysis period post-pandemic would provide further insights into long-term effects on team dynamics and home advantage.

## Conclusion

This study demonstrates that the COVID-19 pandemic had a profound impact on both team rankings and home advantage (HA) in the English Premier League. By introducing an extended Bradley-Terry model that accounts for team-specific home and away effects, we provide a more nuanced and interpretable understanding of competitive dynamics during unprecedented disruptions. Our results reveal that HA was significantly reduced during the pandemic, coinciding with spectator restrictions, but gradually recovered as fans returned. Compared to static and dynamic models, the proposed approach better captures abrupt changes and team-specific variations in HA. These findings highlight the critical role of crowd presence and suggest that home advantage is not uniform across teams or periods of instability. The proposed framework offers a valuable tool for analyzing sports competitions facing external shocks and lays the groundwork for future research across leagues and contexts.

## Acknowledgments

The authors thank the editor and two reviewers for their insightful and constructive comments to improve the paper.

## Author contributions

**Conceptualization:** Hong Hong Liu, Jack Battaglia.

**Formal analysis:** Hong Hong Liu.

**Investigation:** Hong Hong Liu, Jack Battaglia.

**Methodology:** Hong Hong Liu.

**Software:** Hong Hong Liu.

**Supervision:** Tong Tong Wu.

**Validation:** Hong Hong Liu.

**Visualization:** Hong Hong Liu.

**Writing – original draft:** Hong Hong Liu, Jack Battaglia.

**Writing – review & editing:** Tong Tong Wu.

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
