## [Decision Letter · Decision Letter 0]

21 May 2025

PONE-D-25-22110Impact of COVID-19 on English Football Premier League Analyzing Rankings and Home-Field Advantage Using Extended Bradley-Terry ModelsPLOS ONE

Dear Dr. Wu,

Thank you for submitting your manuscript to PLOS ONE. After careful consideration, we feel that it has merit but does not fully meet PLOS ONE’s publication criteria as it currently stands. Therefore, we invite you to submit a revised version of the manuscript that addresses the points raised during the review process.

Based on my own reading of the manuscript, I concur with the reviewers that the paper requires substantial revision. In addition to the points already raised in the reviews, I would particularly expect the following aspects to be addressed:

The authors should provide a clear and explicit rationale for the use of the Bradley-Terry model. This includes discussing alternative approaches relevant to the research question, and articulating the specific advantages of the chosen model in this context.A concise yet comprehensive overview of theoretical explanations for the home advantage phenomenon should be included. The work by Leitner et al. (2022, esp. section 2) as well as the sources provided by Reviewer 1 offers valuable insights that could inform this section.Overall, the manuscript's engagement with the existing literature is rather limited. I would urge the authors to situate their study more firmly within the relevant body of research, clearly outlining how it contributes to, and extends, current knowledge. This is especially pertinent in relation to the literature on football match outcome prediction.

We look forward to receiving your revised manuscript.

Kind regards,

Florian Follert

Academic Editor

PLOS ONE

Reviewers' comments:

Reviewer's Responses to Questions

**Comments to the Author**

1. Is the manuscript technically sound, and do the data support the conclusions?

Reviewer #1: Yes

Reviewer #2: Partly

2. Has the statistical analysis been performed appropriately and rigorously? 

Reviewer #1: Yes

Reviewer #2: No

3. Have the authors made all data underlying the findings in their manuscript fully available?

Reviewer #1: Yes

Reviewer #2: Yes

4. Is the manuscript presented in an intelligible fashion and written in standard English?

Reviewer #1: Yes

Reviewer #2: Yes

5. Review Comments to the Author

Reviewer #1: Thank you for providing me with the opportunity to review your paper.

It analyses the impact of COVID-19 on rankings and home advantage in the English Premier League (EPL) in men’s football.

Its methodological originality is to use extended Bradley-Terry models based on distinct home advantage factors across teams.

I believe the manuscript has potential for publication in the journal.

I have several comments for your attention below.

I hope you find my feedback fair and helpful.

I wish you all the best with further development of your research.

Comments:

In the second sentence of the abstract, the reference to ‘unpredictable match outcomes’ is unsupported and does not look essential to me, I suggest removing it.

The same applies later to your main text lines 10-11.

In your paper, you use only 11 references.

Relatedly, some claims/sentences lack evidence.

For example, the first two sentences need referencing, Butler and Butler (2023) https://www.sciencedirect.com/science/article/pii/S2773161823000058 may be helpful here.

In the first sentence, you refer to the EPL as ‘one of the most prestigious soccer leagues in the world’, it is probably THE most prestigious one.

You also refer to its high level of competition.

However, Scelles et al. (2022) https://www.tandfonline.com/doi/full/10.1080/23750472.2020.1784036 found that its average ‘end of season’ competitive balance over 2006-2018 was average compared to other European first tiers in men’s football.

Therefore, you may want to be more explicit about what you refer to through ‘high level of competition’, while acknowledging contrasting evidence in the literature.

HFA is not explicitly introduced as acronym for home-field advantage until line 49 despite being used earlier.

I would probably argue that home advantage (HA) is more commonly used in the literature.

Line 15, it should be Leitner et al.

The references used in your first paragraph are all supportive of no fans limiting home advantage.

However, Scelles et al. (2024) https://www.tandfonline.com/doi/full/10.1080/19406940.2024.2323012 do not support it.

Therefore, you may want to be more nuanced.

The literature on predicting the outcome of football games lines 26-48 is quite dated.

I suggest updating it, for example by checking more recent references having cited the papers you refer to.

The same applies to the reference to Pollard line 52.

In this specific paragraph about home advantage (that is, lines 48-56), I would expect more discussion about the factors explaining it as per the literature.

This would help better discuss your later findings.

The idea that Manchester United is known for its strong home advantage lines 133-135 needs to be supported.

Is it true since 2013, that is, since Sir Alex Ferguson stopped as head coach?

In other words, was it an Alex Ferguson effect rather than a Manchester United effect?

When presenting the data lines 152-169, I would suggest referring to the introduction of the Video Assistant Referee (VAR) in the league ahead of the 2019-2020 season, see for example Hamsund and Scelles (2021) https://www.mdpi.com/1911-8074/14/12/573 and Teixeira da Silva et al. (2024) https://link.springer.com/article/10.1007/s12283-024-00459-3.

This would help better discuss your findings, for example when referring to ‘the referee’s preference towards the home team’ line 191.

The content lines 174-183 needs to be supported by references.

The same applies to ‘Manchester United, Arsenal, and Chelsea, which had high HAF ranks in pre-COVID due to strong fan bases’ lines 255-257, as this suggests that fan bases alone explain home advantage for these three clubs, while there may be other factors as per the literature on home advantage.

Lines 355-356, a p-value of 0.1133 cannot be deemed marginally significant since it is above 0.1; instead, it should be deemed not significant but not far from being significant at the 10% level.

The discussion section can be improved in several ways:

- referring back to previous literature

- better highlighting your contributions to the literature

- more explicitly identifying the limitations of your study, maybe by grouping them together with directions for future research in a dedicated subsection.

A short conclusion summarising your key findings and contributions could be added.

I do not understand why two authors are thanked in the acknowledgements; the first two sentences should be removed here.

Reviewer #2: The paper deals with an interesting issue. However, the following aspects deserve attention and should be implemented:

1) The fairly simple Bradley-Terry model is extended by one or two parameters. So there is no major innovation in the model.

2) It is not clear why the Bradley-Terry model or an extension of it is used. Why are the Elo model or the Davidson model, in which a draw is already modelled, or even machine learning models such as RankNet etc. not taken into consideration? It would be useful to clearly explain this and justify why the model in question was selected.

3) A brief (!) overview of the potential explanations of the home advantage should be given.

4) It is not clear where the data for the estimation/simulation of µ and β in (4) and (5) come from. It is also not entirely clear whether this is an estimate or a simulation. In the case of the former, a regression output should actually be offered.

5) A presentation of the estimation equations would also be desirable.

6) The new model is a good representation of the situation in the Premier League in the period under consideration. It seems to me that the fitting of this model is given precisely to this league and the period described. In this respect, the question of external validity arises. What does the model say for other time periods and what does it say for other leagues? If it is only suitable for the situation described, then the findings are rather modest.

7) The empirical study is based on data from the 2014 to 2023 seasons. During this period, the personnel in all the clubs analysed (players, managers) were presumably at least partially replaced. How is this dealt with in the study? Apparently, clubs are viewed as anthropomorphic units, and this problem is not addressed. Since personnel policy has a considerable influence on the probability of winning, this should mean that the model considered can only make valid statements for the respective situation and the respective league.

All in all, I don't think the paper is very innovative.

6. PLOS authors have the option to publish the peer review history of their article (what does this mean?). If published, this will include your full peer review and any attached files.

Reviewer #1: **Yes: **Nicolas Scelles

Reviewer #2: No

---

## [Author Response · Author response to Decision Letter 1]

4 Aug 2025

Response to the Editor

Based on my own reading of the manuscript, I concur with the reviewers that the paper requires substantial revision. In addition to the points already raised in the reviews, I would particularly expect the following aspects to be addressed:

We sincerely thank the Editor for the opportunity to revise the paper and insightful and constructive comments. Below, we address each comment raised, noting our revisions and providing relevant citations as requested.

The authors should provide a clear and explicit rationale for the use of the Bradley-Terry model. This includes discussing alternative approaches relevant to the research question, and articulating the specific advantages of the chosen model in this context.

Thank you for pointing out the need for a clearer rationale. We have expanded lines 69–86 to provide a comprehensive discussion of alternative modeling approaches, including Elo-type models [23–25], Davidson models [27], and machine learning-based ranking methods [21]. We now explicitly highlight the advantages of the Bradley-Terry (BT) model: enhanced interpretability, avoidance of path dependence, facilitation of likelihood-based inference and confidence intervals, and transparent handling of ternary outcomes and team-specific parameters.

References cited: [17–26] (see lines 69–86 of the revised manuscript).

A concise yet comprehensive overview of theoretical explanations for the home advantage phenomenon should be included. The work by Leitner et al. (2022, esp. section 2) as well as the sources provided by Reviewer 1 offers valuable insights that could inform this section.

We appreciate this suggestion. Lines 49–55 now contain an expanded overview of theoretical mechanisms for home advantage (HA), citing Leitner et al. [4], Pollard [2], Nevill and Holder [17], Sánchez & García-de-Alcaraz [14], and Bray et al. [15]. We discuss psychological, physiological, and environmental factors and incorporate empirical evidence confirming the strength of HA in football.

References cited: [2, 4, 13–17] (see lines 49–55 of the revised manuscript).

Overall, the manuscript's engagement with the existing literature is rather limited. I would urge the authors to situate their study more firmly within the relevant body of research, clearly outlining how it contributes to, and extends, current knowledge. This is especially pertinent in relation to the literature on football match outcome prediction.

Thank you for this valuable feedback. We have substantially expanded the literature review (lines 56–77) to incorporate more than ten recent works from 2019–2024, covering BT extensions, Elo-based frameworks, deep learning, and Bayesian models in football outcome prediction [18–26]. This additions clarify the context of our contribution, situates our work within current methodological advances, and outlines how our study extends current knowledge, particularly by introducing a team-specific HA parameterization and providing a comparative modeling framework for the COVID-19 disruption.

References cited: [18–26] (see lines 56–77 of the revised manuscript).

Response to the Reviewers

Reviewer #1:

We thank Reviewer #1 for the very helpful suggestions and comments. We have addressed all the points raised and hope that this revision resolves all the concerns.

Thank you for providing me with the opportunity to review your paper. It analyses the impact of COVID-19 on rankings and home advantage in the English Premier League (EPL) in men’s football. Its methodological originality is to use extended Bradley-Terry models based on distinct home advantage factors across teams. I believe the manuscript has potential for publication in the journal. I have several comments for your attention below. I hope you find my feedback fair and helpful. I wish you all the best with further development of your research.

Comments: In the second sentence of the abstract, the reference to ‘unpredictable match outcomes’ is unsupported and does not look essential to me, I suggest removing it. The same applies later to your main text lines 10-11.

Thank you for your thoughtful suggestions on the introduction and referencing. In response, we have removed the unsupported claim regarding “unpredictable match outcomes” from both the abstract and the main text.

In your paper, you use only 11 references. Relatedly, some claims/sentences lack evidence. For example, the first two sentences need referencing, Butler and Butler (2023) https://www.sciencedirect.com/science/article/pii/S2773161823000058 may be helpful here. In the first sentence, you refer to the EPL as ‘one of the most prestigious soccer leagues in the world’, it is probably THE most prestigious one. You also refer to its high level of competition. However, Scelles et al. (2022) https://www.tandfonline.com/doi/full/10.1080/23750472.2020.1784036 found that its average ‘end of season’ competitive balance over 2006-2018 was average compared to other European first tiers in men’s football. Therefore, you may want to be more explicit about what you refer to through ‘high level of competition’, while acknowledging contrasting evidence in the literature.

We have significantly expanded and updated our literature review, now citing more than 30 relevant studies. We have updated the first two sentences of the introduction (lines 2–4) to reference Butler and Butler (2023) [1], which supports the EPL’s status as the most prestigious soccer league and provides context on its financial scale and competitive environment. To avoid overstatement, we removed our earlier claim about competitive balance instead of replacing it with unsubstantiated detail.

HFA is not explicitly introduced as acronym for home-field advantage until line 49 despite being used earlier. I would probably argue that home advantage (HA) is more commonly used in the literature.

The acronym “home advantage (HA)” is now clearly introduced at its first mention (line 4), and we use “HA” consistently throughout the manuscript, in line with many recent studies.

Line 15, it should be Leitner et al.

The citation on line 15 has been corrected to “Leitner et al.” as suggested.

The references used in your first paragraph are all supportive of no fans limiting home advantage. However, Scelles et al. (2024) https://www.tandfonline.com/doi/full/10.1080/19406940.2024.2323012 do not support it. Therefore, you may want to be more nuanced.

We have also incorporated a more nuanced discussion of the impact of fans on home advantage by including recent findings from Scelles et al. (2024) [7] (see lines 21-22).

The literature on predicting the outcome of football games lines 26-48 is quite dated. I suggest updating it, for example by checking more recent references having cited the papers you refer to. The same applies to the reference to Pollard line 52.

Following your recommendation, we have significantly expanded our reference list and now cite over 30 papers, including recent works such as Butler and Butler (2023) and Scelles et al. (2024), to ensure all factual statements are thoroughly supported.

In this specific paragraph about home advantage (that is, lines 48-56), I would expect more discussion about the factors explaining it as per the literature. This would help better discuss your later findings.

Thank you for this suggestion. In the revised manuscript (lines 49–55), we have expanded the discussion of home advantage (HA) mechanisms, explicitly referencing multiple strands of the literature. We now explain that HA is attributed to a combination of factors, including crowd influence on player motivation and referee decisions (Sánchez & García-de-Alcaraz, 2021 [14]; Bray et al., 2003) [15], travel fatigue for away teams (Pollard, 2008 [16]), and venue familiarity (Pollard, 1986 [2]). We also draw on Leitner et al. [4] (2022) for a comprehensive socio-psychological and environmental synthesis, highlight Pollard’s analyses [2] which empirically confirm that football has one of the largest HA effects among team sports, and note Nevill and Holder’s influential review [17] that summarizes these mechanisms.

The idea that Manchester United is known for its strong home advantage lines 133-135 needs to be supported. Is it true since 2013, that is, since Sir Alex Ferguson stopped as head coach? In other words, was it an Alex Ferguson effect rather than a Manchester United effect?

Thank you for the suggestion and we now adopt a more rigorous, data-driven justification. In lines 189–192 of the revised manuscript, we now directly include the supporting data to this point. Specifically, we state that Manchester United rank as the second-strongest home team in Premier League history, winning 61.43% of their home fixtures since 1892/93 [29].

When presenting the data lines 152-169, I would suggest referring to the introduction of the Video Assistant Referee (VAR) in the league ahead of the 2019-2020 season, see for example Hamsund and Scelles (2021) https://www.mdpi.com/1911-8074/14/12/573 and Teixeira da Silva et al. (2024) https://link.springer.com/article/10.1007/s12283-024-00459-3. This would help better discuss your findings, for example when referring to ‘the referee’s preference towards the home team’ line 191.

Thank you for this valuable suggestion. In the revised manuscript, we have incorporated a discussion of the introduction of the Video Assistant Referee (VAR) in the EPL before the 2019–2020 season. We have added references to Hamsund and Scelles (2021) and Teixeira da Silva et al. (2024) in lines 218–220 to highlight VAR’s impact on match outcomes and its potential to mitigate home-team refereeing bias.

The content lines 174-183 needs to be supported by references. The same applies to ‘Manchester United, Arsenal, and Chelsea, which had high HAF ranks in pre-COVID due to strong fan bases’ lines 255-257, as this suggests that fan bases alone explain home advantage for these three clubs, while there may be other factors as per the literature on home advantage.

Thank you for emphasizing the need for stronger evidence. In the revised manuscript, we have added relevant references to substantiate our claims.

For the statements regarding Manchester United, Arsenal, and Chelsea and their fan bases (now lines 315–318), we have cited empirical attendance data from Sky Sports (“Premier League average attendances by club 2023”: https://www.skysports.com/football/news/11095/12896864/future-of-football-attendances-on-the-rise-but-how-big-could-stadiums-act)[35], which documents the historically high average home attendances for these clubs. This data provides concrete evidence of their large and consistent supporter bases.

Regarding Arsenal’s post-COVID recovery (lines 237–243), we now cite James, J. (2022, May 23)[32] “Our 2021/22 season in numbers” and James, J. (2023, May 31)[33] “Games, goals and records: 2022/23 in numbers” from Arsenal.com, along with McGuigan, M. (2017)[34] “Monitoring training and performance in athletes” (Human Kinetics), to support our discussion that the club’s youthful squad may have aided faster physical recovery after the pandemic disruption.

Lines 355-356, a p-value of 0.1133 cannot be deemed marginally significant since it is above 0.1; instead, it should be deemed not significant but not far from being significant at the 10% level.

The language around p-values has been corrected. We now refer to results as “not significant but not far from being significant at the 10% level” where appropriate.

The discussion section can be improved in several ways: - referring back to previous literature - better highlighting your contributions to the literature - more explicitly identifying the limitations of your study, maybe by grouping them together with directions for future research in a dedicated subsection. A short conclusion summarising your key findings and contributions could be added.

We have enhanced the Discussion section by highlighting our contributions and explicitly outlining limitations and directions for future research in a dedicated subsection (see lines 520-525). A concise Conclusion summarizing key findings of the study is now added (see lines 536-538).

I do not understand why two authors are thanked in the acknowledgements; the first two sentences should be removed here.

Thank you for catching that. The acknowledgements section has been revised as suggested.

Reviewer # 2:

The paper deals with an interesting issue. However, the following aspects deserve attention and should be implemented:

We thank Reviewer #2 for the insightful suggestions and comments. We have carefully addressed each point raised and hope that this revision adequately meets all the concerns.

1) The fairly simple Bradley-Terry model is extended by one or two parameters. So there is no major innovation in the model.

Our model extends the Bradley–Terry framework with several novel features and is now clearly articulated in the manuscript. First, unlike prior BT models, our proposed framework introduces distinct home and away ability parameters for each team. This enables team-specific estimates of HA, for example, distinguishing Manchester United’s and Burnley’s relative home strengths even if their overall abilities are comparable. Second, by omitting match-level covariates and team-specific intercepts, we avoid the need for penalization and achieve a more parsimonious and interpretable model that still captures the essential variation in match outcomes. Third, this streamlined structure provides greater robustness during periods of abrupt structural shifts (e.g., COVID-19 fan absence), yielding improved parameter stability and better predictive fit across pre- and mid-pandemic periods. Finally, in contrast to many previous extended BT models, which often rely on team-specific random walks or Bayesian updating, our approach offers a conceptually simpler estimation procedure with fewer assumptions while still capturing temporal dynamics in team performance and home advantage across pre-pandemic, pandemic, and post-pandemic periods.

In addition to the model extension, we apply the Bradley–Terry framework to assess the impact of COVID-19 on football via home advantage, illustrating how this approach can be adapted for other sports and disruptive events.

2) It is not clear why the Bradley-Terry model or an extension of it is used. Why are the Elo model or the Davidson model, in which a draw is already modelled, or even machine learning models such as RankNet etc. not taken into consideration? It would be useful to clearly explain this and justify why the model in question was selected.

We added three new paragraphs in the Introduction (lines 56-92) explaining why the BT model was chosen over the Elo, Davidson, and RankNet approaches. The main advantages of the BT model include: (1) Explicit ternary outcome modeling: The BT model uses an ordered logistic regression to handle ternary outcomes (win/draw/loss) directly, whereas Elo methods typically omit draws. Szczecinski et al. (2022) showed that FIFA’s Elo ranking system approximates an online Bradley–Terry model, and advocated for a more data-driven, hybrid alternative. Honda (2024) further argued that the Bradley–Terry (BT) model yields more stable and interpretable strength estimates than Elo, as it avoids path dependence and considers the full history of match results. (2) Flexible HA structure: While Davidson models allow for draws, they do not accommodate team-specific HA parameterization our study requires; (3) Interpretability and data efficiency: Machine-learning ranking models (e.g., RankNet) demand large datasets and sacrifice interpretability, making them unsuitable for our relatively small COVID-period data; (4) Transparency and simplicity: The BT framework facilitates both inferential transparency and computational simplicity, making it well-suited for assessing HA dynamics during disrupted seasons. The BT model allows for likelihood-based inference and confidence intervals, which are typically unavailable in standard Elo frameworks. More details can be found in the main paper.

3) A brief (!) overview of the potential explanations of the home advantage should be given.

We now provide a theoretical overview of home advantage mechanisms in the Introduction (lines 49-55), grounded in rec

---

## [Decision Letter · Decision Letter 1]

2 Sep 2025

Impact of COVID-19 on English Football Premier League Analyzing Rankings and Home Advantage Using Extended Bradley-Terry Models

PONE-D-25-22110R1

Dear Dr. Wu,

We’re pleased to inform you that your manuscript has been judged scientifically suitable for publication and will be formally accepted for publication once it meets all outstanding technical requirements.

Kind regards,

Prof. Dr. Florian Follert

Academic Editor

PLOS ONE

---

## [Editor Report · Acceptance letter]

PONE-D-25-22110R1

PLOS ONE

Dear Dr. Wu,

I'm pleased to inform you that your manuscript has been deemed suitable for publication in PLOS ONE. Congratulations! Your manuscript is now being handed over to our production team.

Kind regards,

on behalf of

Prof. Dr. Florian Follert

Academic Editor

PLOS ONE